# Efficient ensemble data assimilation for coupled models with the Parallel Data Assimilation Framework: Example of AWI-CM (AWI-CM-PDAF 1.0)

Lars Nerger[1], Qi Tang[1], and Longjiang Mu[1]

[1]Alfred-Wegener-Institut Helmholtz-Zentrum für Polar- und Meeresforschung, Bremerhaven, Germany

**Correspondence:** Lars Nerger (lars.nerger@awi.de)

**Abstract.** Data assimilation integrates information from observational measurements with numerical models. When used with coupled models of Earth system compartments, e.g. the atmosphere and the ocean, consistent joint states can be estimated. A common approach for data assimilation are ensemble-based methods which utilize an ensemble of state realizations to estimate the state and its uncertainty. These methods are far more costly to compute than a single coupled model because of the required integration of the ensemble. However, with uncoupled models, the ensemble methods also have been shown to exhibit a particularly good scaling behavior. This study discusses an approach to augment a coupled model with data assimilation functionality provided by the Parallel Data Assimilation Framework (PDAF). Using only minimal changes in the codes of the different compartment models, a particularly efficient data assimilation system is generated that utilizes parallelization and in-memory data transfers between the models and the data assimilation functions and hence avoids most of the file reading and writing, and also model restarts during the data assimilation process. The study explains the required modifications of the programs on the example of the coupled atmosphere-sea ice-ocean model AWI-CM. Using the case of the assimilation of oceanic observations shows that the data assimilation leads only to small overheads in computing time of about 15% compared to the model without data assimilation and a very good parallel scalability. The model-agnostic structure of the assimilation software ensures a separation of concerns in which the development of data assimilation methods can be separated from the model application.

*Copyright statement.* TEXT

## 1 Introduction

Data assimilation (DA) methods are used to combine observational information with models. A common application is to apply DA to estimate an initial state that is used to start a forecast system as is common practice at weather and marine forecasting centers. The most widely used class of ensemble DA methods are ensemble-based Kalman filters (EnKFs) like the local ensemble transform Kalman filter (LETKF, Hunt et al., 2007), the deterministic ensemble Kalman filter (DEnKF, Sakov and Oke, 2008), or the local error-subspace transform Kalman filter (LESTKF, Nerger et al., 2012b). Commonly, the DA is

applied to separate models simulating e.g. the atmospheric dynamics or the ocean circulation. However, in recent years coupled models of different Earth system compartments have become more common. In this case, the compartment models frequently exchange information at the interface of the model domains to influence the integration of the other model compartment. For example, in coupled atmosphere-ocean models the fluxes through the ocean surface are dynamically computed based on the physical state of both the atmosphere and the ocean are exchanged in between both compartments. For model initialization, DA should be applied to each of the compartments. Here, the DA can either be performed separately in the different compartment domains, commonly called weakly-coupled DA, or it can be performed in a joint update, called strongly-coupled DA. Only strongly coupled DA is expected to provide fully dynamically consistent state estimates.

A recent overview of methods and issues in coupled DA is provided by Penny et al. (2017). By now the weakly coupled assimilation is the common choice for assimilation into coupled models and recent studies assess the effect of this assimilation approach. For atmosphere-ocean coupled models, different studies either assimilated observations of one compartment into the observed compartment (e.g. Kunii et al. (2017); Mu et al. (2020)) or observations of each compartment into the corresponding one (e.g. Zhang et al. (2007); Liu et al. (2013); Han et al. (2013); Chang et al. (2013); Lea et al. (2015); Karspeck et al. (2018); Browne et al. (2019)). The research question considered in these studies is usually to which extent the assimilation into a coupled model can improve predictions in comparison to the assimilation into uncoupled models. Partly, the mentioned studies used twin experiments assimilating synthetic observations to assess the DA behavior.

Strongly coupled DA is a much younger approach, which is not yet well established. Open questions for strongly coupled DA are for example how to account for the different temporal and spatial scales in the atmosphere and the ocean. Strongly coupled DA is complicated by the fact that DA systems for the ocean and atmosphere have usually been developed separately and often use different DA methods. For example, Laloyaux et al. (2016) used a 3D variational DA in the ocean, but 4D variational DA in the atmosphere. The methodology lead to a quasi-strongly coupled DA. Frolov et al. (2016) proposed an interface-solver approach for variational DA methods, which leads to a particular solution for the variables close to the interface. Strongly coupled DA was applied by Sluka et al. (2016) in a twin experiment using an EnKF with dynamically estimated covariances between the atmosphere and ocean in a low-resolution coupled model. For coupled ocean-biogeochemical models, Yu et al. (2018) discussed strongly coupled DA in an idealized configuration. Further, Goodliff et al. (2019) discussed the strongly coupled DA for a coastal ocean-biogeochemical model assimilating real observations of sea surface temperature. This study pointed to the further complication of the choice of variable (linear or logarithmic concentrations for the biogeochemical compartment) for strongly coupled assimilation.

Ensemble-based Kalman filters, but also the nonlinear particle filters, can be formulated to work entirely on state vectors. A state vector is the collection of all model fields at all model grid points in form of a vector. When one computes the observed part of the state vector, applying the so-called 'observation operator', one needs to know how a field is stored in the state vector. However, the core part of the filter, which computes the corrected state vector (the so-called 'analysis state') taking into account the observational information, does not need to know how the state vector is constructed. This property is also important for coupled DA, where the state vector will be distributed over different compartments, like the atmosphere and the ocean.

The possibility to implement most parts of a filter algorithm in a generic model-agnostic way has motivated the implementation of software frameworks for ensemble DA. While the frameworks use very similar filter methods, they differ strongly in the strategy how the coupling between model and DA software is achieved. As described by Nerger et al. (2012b) one can distinguish between offline and online DA coupling. In offline-coupled DA one uses separate programs for the model and the assimilation and performs the data transfer between both through disk files. In online coupled DA one performs in-memory data transfer, usually by parallel communication, and hence avoids the use of disk files. In addition, online-couped DA avoids the need to stop and restart a model for the DA. The Data Assimilation Research Testbed (DART, Anderson et al., 2009) uses file transfers and separate programs for the ensemble integration and the filter analysis step, which are run consecutively. The framework 'Employing Message Passing Interface for Researching Ensembles' (EMPIRE, Browne and Wilson, 2015) uses parallel communication between separate programs for model and DA. These programs are run in parallel and the information transfer is performed through the parallel communication, which avoids data transfers using files. The Parallel Data Assimilation Framework (PDAF, Nerger et al., 2005, 2012b, http://pdaf.awi.de) supports both online and offline coupled DA. For the online coupled DA, PDAF also uses parallel communication. However, in contrast to EMPIRE, the model usually is augmented by the DA functionality, i.e., model and DA are compiled into a joint program.

For coupled ensemble DA in hydrology, Kurtz et al. (2016) have combined PDAF with the coupled terrestrial model system TerrSysMP. To build the system, a wrapper was developed to perform the online-coupling of model and DA software. The study shows that the resulting assimilation system is highly scalable and efficient. Karspeck et al. (2018) have discussed a coupled atmosphere-ocean DA system. They apply the DART software and perform weakly coupled DA using two separate ensemble-based filters for the ocean and atmosphere, which produce restart files for each model compartment. These are then used to initialize the ensemble integration of the coupled model.

Here, we discuss a strategy to build an online-coupled DA system for coupled models on the example of the coupled atmosphere-ocean model AWI-CM. The strategy enhances the one discussed in Nerger et al. (2012b) for an ocean-only model. The previous strategy is modified for the coupled DA and applied to the two separate programs for the atmosphere and ocean, which together build the coupled model AWI-CM (Sidorenko et al., 2015). The required modifications to the model source codes consist essentially in adding four subroutine calls in each of the two compartment models. Three of these subroutine calls connect the models to the DA functionality provided by PDAF, while the fourth is optional and provides timing and memory information. With this strategy, a wrapper that combines the compartment model into a single executable as used by Kurtz et al. (2016) can be avoided. We discuss the strategy for both weakly and strongly coupled DA but assess the parallel performance only for weakly coupled DA into the ocean, which is supported in the code version AWI-CM-PDAF V1.0. This is motivated by the fact that strongly coupled DA is not yet well established and weakly coupled DA by itself is a topic of current research.

The remainder of the study is structured as follows: Section 2 discusses ensemble filters and their setup for coupled DA. The setup of a DA system is described in Section 3. Section 4 discusses the parallel performance of the DA system build by coupling AWI-CM and PDAF. Section 5 examines the assimilation behavior of an example application with AWI-CM. Implications of the chosen strategy to coupled the model and data assimilation are discussed in Sec. 6. Finally, conclusions are drawn in Sec. 7.

## 2 Ensemble filters

Ensemble DA (EnDA) methods use an ensemble of model state realizations to represent the state estimate (usually the ensemble mean) and the uncertainty of this estimate given by the ensemble spread. The filters perform two alternating phases: In the forecast phase the ensemble of model states is integrated with the numerical model until the time when observations are available. At this time, the analysis step is computed. It combines the information from the model state and the observations taking into account the estimated error of the two information sources and computes an updated model state ensemble, which represents the analysis state estimate and its uncertainty.

The currently most widely used ensemble filter methods are ensemble-based Kalman filters based on the Ensemble Kalman filter (Evensen, 1994; Houtekamer and Mitchell, 1998; Burgers et al., 1998). When incorporating the observations during the analysis step, these filters assume that the errors in the state and the observations are Gaussian distributed. This allows to formulate the analysis step just using the two leading moments of the distributions, namely the mean and covariance matrix. Another class of EnDA methods are particle filters (e.g., van Leeuwen, 2009). While particle filters do not assume Gaussianity of error distributions, they are difficult to use with high-dimensional models because particular adaptions are required to avoid that the ensemble collapses to a single member due to the so-called 'curse of dimensionality' (see Snyder et al., 2008). Methods to make particle filters usable for high-dimension systems were reviewed by van Leeuwen et al. (2019). One strategy is to use the observational information already during the forecast phase to keep the ensemble states close to the observations. This approach requires that some DA functions are already executed during the forecast phase. The realization in the implementation strategy will be discussed in Sec. 3.2.

### 2.1 Filter algorithms

To be able to discuss the particularities of coupled DA with respect to ensemble filter, here the error-subspace transform Kalman filter (ESTKF, Nerger et al., 2012b) is reviewed. The ESTKF is an efficient formulation of the EnKF that has been applied in different studies to assimilate satellite data into sea-ice ocean models (e.g. Kirchgessner et al., 2017; Mu et al., 2018; Androsov et al., 2019) and biogeochemical ocean models (e.g. Pradhan et al., 2019; Goodliff et al., 2019).

#### 2.1.1 The ESTKF

In the analysis step at the time $t_k$, the ESTKF transforms a forecast ensemble $\mathbf{X}_k^f$ of $N_e$ model states of size $N_x$ stored in the columns of this matrix into a matrix of analysis states $\mathbf{X}_k^a$ as

$$\mathbf{X}_k^a = \overline{\mathbf{x}}_k^f \mathbf{1}_{N_e}^T + \mathbf{X}_k^f \left( \mathbf{w}_k \mathbf{1}_{N_e}^T + \tilde{\mathbf{W}}_{\mathbf{k}} \right) \tag{1}$$

where $\overline{\mathbf{x}}_k^f$ is the forecast ensemble mean state and $\mathbf{1}_{N_e}$ is a vector of size $N_e$ holding the value one in all elements. Further, $\mathbf{w}_k$ is a vector of size $N_e$ which transforms the ensemble mean and $\tilde{\mathbf{W}}$ is a matrix of size $N_e \times N_e$ which transforms the ensemble perturbations. Below, the time index $k$ is omitted, as all computations in the analysis refer to the time $t_k$.

The forecast ensemble represents an error-subspace of dimension $N_e - 1$ and the ESTKF computes the ensemble transformation matrix and vector in this subspace. Practically, one computes an error-subspace matrix by $\mathbf{L} = \mathbf{X}^f \mathbf{T}$ where $\mathbf{T}$ is a projection matrix with $j = N_e$ rows and $i = N_e - 1$ columns defined by

$$\mathbf{T}_{j,i} = \begin{cases} 1 - \frac{1}{N_e} \frac{1}{\frac{1}{\sqrt{N_e}}+1} & \text{for } i = j, j < N_e \\ -\frac{1}{N_e} \frac{1}{\frac{1}{\sqrt{N_e}}+1} & \text{for } i \neq j, j < N_e \\ -\frac{1}{\sqrt{N_e}} & \text{for } j = N_e. \end{cases} \tag{2}$$

Below, the equations are written using $\mathbf{X}^f$ and $\mathbf{T}$ rather than $\mathbf{L}$ as this leads to a more efficient formulation.

A model state vector $\mathbf{x}^f$ and the vector of observations $\mathbf{y}$ with dimension $N_y$ are related through the observation operator $\mathbf{H}$ by

$$\mathbf{y} = \mathbf{H}\left(\mathbf{x}^f\right) + \epsilon \tag{3}$$

where $\epsilon$ is the vector of observation errors, which are assumed to be a white Gaussian distributed random process with observation error covariance matrix $\mathbf{R}$. For the analysis step, a transform matrix in the error-subspace is computed as

$$\mathbf{A}^{-1} = \rho(N_e - 1)\mathbf{I} + (\mathbf{H}\mathbf{X}^f\mathbf{T})^T \mathbf{R}^{-1} \mathbf{H}\mathbf{X}^f\mathbf{T} . \tag{4}$$

This matrix provides ensemble weights in the error-subspace. The factor $\rho$ with $0 < \rho \leq 1$ is called the "forgetting factor" (Pham et al., 1998) and is used to inflate the forecast error covariance matrix. The weight vector $\mathbf{w}_k$ and matrix $\tilde{\mathbf{W}}$ are now given by

$$\mathbf{w} = \mathbf{T}\mathbf{A}\left(\mathbf{H}\mathbf{X}^f\mathbf{T}\right)^T \mathbf{R}^{-1}\left(\mathbf{y} - \mathbf{H}\bar{\mathbf{x}}^f\right) , \tag{5}$$

$$\tilde{\mathbf{W}} = \sqrt{N_e - 1}\,\mathbf{T}\mathbf{A}^{1/2}\mathbf{T}^T \tag{6}$$

where $\mathbf{A}^{1/2}$ is the symmetric square root which is computed from the eigenvalue decomposition $\mathbf{U}\mathbf{S}\mathbf{U}^T = \mathbf{A}^{-1}$ such that $\mathbf{A}^{1/2} = \mathbf{U}\mathbf{S}^{-1/2}\mathbf{U}^T$. Likewise, $\mathbf{A}$ in Eq. (5) is computed as $\mathbf{A} = \mathbf{U}\mathbf{S}^{-1}\mathbf{U}^T$.

For high-dimensional models a localized analysis is computed following Nerger et al. (2006). Here, each vertical column of the model grid is updated independently by a local analysis step. For updating a column, only observations within a horizontal influence radius $l$ are taken into account. Thus, the observation operator is local and computes an observation vector within the influence radius $l$ from the global model state. Further, each observation is weighted according to its distance from the water column to down-weight observations at larger distances (Hunt et al., 2007). The weight is applied by modifying matrix $\mathbf{R}^{-1}$ in Eqns. (4) and (5). The localization weight for the observations is computed from a correlation function with compact support given by a 5th-order polynomial with a shape similar to a Gaussian function (Gaspari and Cohn, 1999). The localization leads to individual transformation weights $\mathbf{w}_k$ and $\tilde{\mathbf{W}}$ for each local analysis domain.

## 2.2 Weakly-coupled ensemble filtering

In weakly coupled DA, the EnKF is applied in the coupled model to a single compartment or separately to several of the compartments. Given that the analysis is separate for each involved compartment, the filter is applied as in a single-compartment

model. Thus, in practice several EnKFs compute the analyses updates independently before the next forecast phase is started with the updated fields from the different compartments.

## 2.3 Strongly-coupled ensemble filtering

To discuss strongly-coupled filtering, let us assume a two-compartment system (perhaps the atmosphere and the ocean). Let
$\mathbf{x}_A$ and $\mathbf{x}_O$ denote the separate state vector in each compartment. For strongly-coupled DA, both are joined into a single state vector $\mathbf{x}_C$.

Using the joint forecast ensemble $\mathbf{X}_C^f$ in Eq. (1) of the ESTKF one sees that the same ensemble weights $\mathbf{w}, \tilde{\mathbf{W}}$ are applied to both $\mathbf{x}_A$ and $\mathbf{x}_O$. The weights are computed using Eqns. (4) to (6). These equations involve the observed ensemble $\mathbf{H}\mathbf{X}_C^f$, the observation vector $\mathbf{y}$, and the observation error covariance matrix $\mathbf{R}$. Thus, for strongly coupled DA, the updated weights
depend on which compartment is observed. If there are observations of both compartments they are jointly used to compute the weights. If only one compartment is observed, e.g having only ocean observations $\mathbf{y}_O$, then we also have $\mathbf{H}\mathbf{X}_C^f = (\mathbf{H}\mathbf{X}^f)_O$ and the weights are only computed from these observations. Thus, through Eq. (1), the algorithm can directly update both compartments $\mathbf{x}_A$ and $\mathbf{x}_O$ using observations of just one compartment.

An interesting aspect is that when one runs separate assimilation systems for the two compartments with the same filter
methodology, one can compute a strongly-coupled analysis by only exchanging the parts of $\mathbf{y}$, $\mathbf{H}\mathbf{X}^f$, and $\mathbf{R}$ in between both compartments and then initializing the vectors containing observational information from all compartments in the assimilation system of each compartment. If there are only observations in one of the compartments, one can also compute the weights in that compartment and provide them to the other compartment. Given that $\mathbf{y}$ and $\mathbf{R}$ are initialized from information that is usually stored in files, one can also let the DA code coupled into each compartment model read these data and only exchange
the necessary parts of $\mathbf{H}\mathbf{X}^f$. While this discussion shows that technically it is straightforward to apply strongly-coupled DA with these filter methods, one has to account for the model parallelization, which is discussed in Section 3.3.

## 3 Setup of data assimilation program

This section describes the assimilation framework and the setup of the DA program. First an overview of PDAF is given (Sec. 3.1). The code modifications for online-coupling are described in Sec. 3.2, the modifications of the parallelization are described
in Sec. 3.3. Finally, Sec. 3.4 explains the aspect of the call-back functions.

The setup builds on the strategy introduced by Nerger and Hiller (2013). Here, the discussion focuses on the particularities when using a coupled model consisting of separate executable programs for each compartment. While we here describe both the features for weakly and strongly coupled DA, AWI-CM-PDAF in version 1.0 is only coded with weakly-coupled DA into the ocean. This is motivated by the fact that the weakly-coupled DA into a coupled climate model has already different
properties than DA in an uncoupled model. In particular, the initial errors in the coupled AWI-CM are much larger than in a simulation of FESOM using atmospheric forcing. Mainly this is because in FESOM the forcing introduces information about the weather conditions, while AWI-CM only represents the climate state. Thus studying weakly-coupled DA, which is still used

in most applications, has a value on its own. Strongly coupled DA will be supported in the AWI-CM-PDAF model binding in the future.

## 3.1 Parallel Data Assimilation Framework (PDAF)

PDAF (Nerger and Hiller, 2013, http://pdaf.awi.de) is free open-source software that was developed to simplify the implementation and application of ensemble DA methods. PDAF provides a generic framework containing fully implemented and parallelized ensemble filter and smoother algorithms like the LETKF (Hunt et al., 2007), the ESTKF (Nerger et al., 2012b), or the nonlinear NETF method (Tödter and Ahrens, 2015) and related smoothers (e.g., Nerger et al., 2014; Kirchgessner et al., 2017). Further, it provides functionality to adapt a model parallelization for parallel ensemble forecasts as well as routines for the parallel communication linking the model and filters. Analogous to many large-scale geoscientific simulation models, PDAF is implemented in Fortran and is parallelized using the Message Passing Interface standard (MPI, Gropp et al., 1994) as well as OpenMP (OpenMP, 2008). This ensures optimal compatibility with these models, while it is still usable with models coded, e.g., in the programming language C.

The filter methods are model-agnostic and only operate on abstract state vectors as described for the ESTKF in Sec. 2. This allows to develop the DA methods independently from the model and to easily switch between different assimilation methods. Any operations specific to the model fields, the model grid, or to the assimilated observations are performed in program routines provided by the user based on existing template routines. The routines have a specified interface and are called by PDAF as call-back routines, i.e. the model code calls routines of PDAF, which then call the user routines. This call structure is sketched in Fig. 1. Here, an additional yellow 'interface routine' is used in between the model code and the PDAF library routine. This interface routine is used to define parameters for the call to the PDAF library routines, so that these do not need to be specified in the model code. Thus, only a single-line call to each interface routine is added to the model code, which keeps the changes to the model code to a minimum.

The motivation for this call structure is that the call-back routines exist in the context of the model (i.e. the user space) and can be implemented like model routines. In addition, the call-back routines can access static arrays allocated by the model, e.g. through Fortran modules or C header files. For example, this can be used to access arrays holding model fields or grid information. This structure can also be used in case of an offline-coupling using separate programs for the model and the analysis step. However, in this case the grid information is not already initialized by the model and has to be initialized by a separate routine. Using the interfaces and user routines provided by PDAF, it can also be used with models implemented in C or C++, or can be combined with Python. For coupled models consisting of multiple executables, this call structure is used for each compartment model.

## 3.2 Augmenting a coupled model for ensemble data assimilation

Here, only the online-coupling for DA is discussed. As described before, the offline-coupling uses separate programs for the model and the DA program and model restart files to transfer information about the model states between both programs. Generally, the same code for the user routines can be used for online and offline coupled DA. The difference is that in the

online coupling, model information like the model grid are initialized by the model code and usually stored in e.g. Fortran modules. For offline coupled DA one could use the same variable names, and the same names for the modules. Thus, one would need to implement routines that initialize these variables.

The strategy to augment a coupled model with DA functionality is exemplified here using the AWI climate model (AWI-CM, Sidorenko et al., 2015). The model consists of the atmospheric model ECHAM6 (Stevens et al., 2013), which includes the land surface model JSBACH, and the finite-element sea-ice ocean model (FESOM, Danilov et al., 2004; Wang et al., 2008). Both models are coupled using the coupler library OASIS3-MCT (Ocean-Atmosphere-Sea-Ice-Soil coupler - Model Coupling Toolkit, Valcke, 2013). OASIS3-MCT computes the fluxes between the ocean and the atmosphere and performs the interpolation between both model grids. The coupled model consists of two separate programs for ECHAM and FESOM, which are jointly started on the computer so that they can exchange data via the Message Passing Interface (MPI, Gropp et al., 1994). OASIS-MCT is linked into each program as a library. For further details on the model, we refer to Sidorenko et al. (2015).

The online coupling for DA was already discussed in Nerger and Hiller (2013) for an earlier version of the ocean model used in the AWI-CM. Here, an updated coupling strategy is discussed that requires less changes to the model code. While the general strategy for online coupling of the DA is the same as in the previous study, we privde here a full description for completeness. Further, we discuss the particularities of the coupled model.

Figure 2 shows the general program flow and the necessary extension of the code for adding the DA functionality. The different boxes can, but are not required to be subroutine calls. The figure is valid for any of the two programs of the coupled model system. Without the references to the coupler it would also be valid for a single-compartment model.

The left hand side of Fig. 2 shows the typical flow of a coupled compartment model. Here, at the very beginning of the program, the parallelization is initialized ('init. parallelization'). After this step, all involved processes of the program are active (for the parallelization aspects see Sec. 3.3). Subsequently, the OASIS coupler initializes the parallelization for the coupled model, by separating the processes for ECHAM and FESOM. Thus, after this point, the coupler can distinguish the different model compartments. Now, the model itself is initialized, e.g. the model grid for each compartment is initialized and the initial fields are read from files. Further, information for the coupling will be initialized like the grid configuration, which is required by the coupler to interpolate data in between the different model grids. This completes the model initialization and the time stepping is computed. During the time stepping, the coupler exchanges the interface information between the different compartments. After the time stepping some post-processing can be performed, e.g. writing time averages or restart files to disk.

The right hand side of Fig. 2 shows the required additions to the model code as yellow boxes. These additions are calls to subroutines that interface between the model code and the DA framework. In this way, only single-line subroutine calls are added, which might be enclosed in preprocessor checks to allow to activate or deactivate the data-assimilation extension at compile time. The additions are done in both the codes of ECHAM and FESOM and here we discuss them in general. The added subroutine calls have the following functionality:

– *Init_parallel_PDAF*: This routine modifies the parallelization of the model. Instead of integrating the state of a single model instance ('model task'), the model is modified to run an ensemble of model tasks. This routine is inserted directly after the parallelization is started. So all subsequent operations of the program will act in the modified parallelization. In the coupled model this routine is executed before the parallelization of the coupler is initialized. In this way also the coupler will be initialized for an ensemble of model states.

– *Init_PDAF*: In this routine the PDAF framework will be initialized. The routine is inserted into the model codes so that it is executed after all normal model initialization is completed; thus just before the time-stepping loop. The routine specifies parameters for the DA, which can be read from a configuration file. Then, the initialization routine for PDAF, named 'PDAF_init' is called, which performs the PDAF-internal configuration and allocates the internal arrays, e.g. the array of the ensemble states. Further, the initial ensemble is read from input files. As this reading is model-specific, it is
performed by a user-provided routine that is called by PDAF as a call-back routine (see Sec. 3.4). After the framework is initialized, the routine 'PDAF_get_state' is called. This routine writes the information from the initial ensemble into the field arrays of the model. In addition, the length of the initial forecast phase, i.e. the number of time steps until the first analysis step, is initialized. For the coupled model, 'PDAF_init' and 'PDAF_get_state' are called in each compartment. However, some parameters are distinct. For example, the time step size of ECHAM if 450s, while it is 900s for FESOM.
Hence, the number of time steps in the forecast phase of one day are different in the compartments.

    – *Assimilate_PDAF*: This routine is called at the end of each model time step. For this, it is inserted into the model codes of ECHAM and FESOM at the end of the time stepping loop. The routine calls a filter-specific routine of PDAF that computes the analysis step of the selected filter method, for example 'PDAF_assimilate_lestkf' for the localized ESTKF. This routine of PDAF also checks whether all time steps of a forecast phase have been computed. Only if this
is true, the analysis step is executed while otherwise the time stepping is continued. If additional operations for the DA are required during the time stepping, like taking into account future observations in case of the advanced equivalent-weights particle filter (EWPF, van Leeuwen, 2010) or collecting observed ensemble fields during the forecast phase for a 4-dimensional filtering (Harlim and Hunt, 2007), these are also performed in this filter-specific routine. For the coupled model, the routine is called in both ECHAM and FESOM. Then, 'PDAF_assimilate_lestkf' will check for the analysis
time according to the individual number of time steps in the forecast phase. The analysis step will then be executed in each compartment according to the configuration of the assimilation. In the implementation of AWI-CM-PDAF 1.0, the analysis is only performed in FESOM. Thus, while 'PDAF_assimilate_lestkf' is also called in ECHAM, is does not assimilate any data.

    – *Finalize_PDAF*: This routine is called at the end of the program. The routine includes calls to the routine 'PDAF_print_info',
which print out information about execution times of different parts of the assimilation program as measured by PDAF as well as information about the memory allocated by PDAF.

Compared to the implementation strategy discussed in Nerger and Hiller (2013), in which the assimilation subroutine is only called after a defined number of time steps, this updated scheme allows to perform DA operations during the time stepping loop. To use this updated scheme, one has to execute the coupled model with enough processors so that all ensemble members can be run at the same time. This is nowadays easier than in the past because the number of processor cores is much larger in current high-performance computers compared to the past.

Apart from the additional subroutine calls, a few changes were required in the source codes of ECHAM, FESOM, and OASIS3-MCT which are related to the parallelization. These changes are discussed in Sec. 3.3.

### 3.3 Parallelization for coupled ensemble data assimilation

The modification of the model parallelization for ensemble DA is a core element of the DA online coupling. Here, the parallelization of AWI-CM and the required changes for the extension for the DA are described. For FESOM, as a single-compartment model, the adaption of the parallelization was described by Nerger et al. (2005) and Nerger and Hiller (2013). A similar parallelization was also described by Browne and Wilson (2015). For the online-coupling of PDAF with the coupled model TerrSysMP, the setup of the parallelization was described by Kurtz et al. (2016). While for TerrSysMP a different coupling strategy was used, the parallelization of the overall system is essentially the same as discussed here for AWI-CM. The parallelization for the DA is configured by the routine *init_parallel_pdaf*. In general this is a template routine, which can be adapted by the user according to the particular needs. Nonetheless, by now the default setup in PDAF was directly usable in all single-compartment models to which PDAF was coupled. Compared to the default setup in PDAF for a single-compartment model, we have adapted the routine to account for the existence of two model compartments.

Like other large-scale models, AWI-CM is parallelized using the Message-Passing Interface standard (MPI, Gropp et al., 1994). MPI allows to compute a program using several processes with distributed memory. Thus, each process has only access to the data arrays that are allocated by this process. Data exchanges between processes are performed in form of parallel communication, i.e. the data is explicitly sent by one process and received by another process. All parallel communication is performed within so-called communicators, which are groups of processes. When the parallel region of a program is initialized, the communicator MPI_COMM_WORLD is initialized, which contains all processes of the program. In case of AWI-CM when the two executables for ECHAM and FESOM are jointly started, they share the same MPI_COMM_WORLD so that parallel communication between the processses running ECHAM and those running FESOM is possible. Further communicators can be defined by splitting MPI_COMM_WORLD. This is used to define groups of processes both for AWI-CM and for the extension with PDAF.

For AWI-CM without data-assimilation extension, the parallelization is initialized by each program at the very beginning. Then, a routine of OASIS-MCT is called which splits MPI_COMM_WORLD into two communicators: one for ECHAM (COMM_ECHAM) and one for FESOM (COMM_FESOM). These communicators are then used in each of the compartment models and together they build one model task that integrates one realization of the coupled model state. MPI_COMM_WORLD is further used to define one process each for ECHAM and FESOM, which perform the parallel communication to exchange flux information. Important is here, that OASIS-MCT is coded to use MPI_COMM_WORLD to define these communicators.

Each of the compartment models then uses its group of processes for all compartment-internal operations. Each model uses a domain-decomposition, i.e. each process computes a small region of the global domain in the atmosphere or the ocean. The distribution of the processes is exemplified in Fig. 3(a) for the case of 6 processes in MPI_COMM_WORLD. Here, the communicator is split into 4 processes for COMM_FESOM (green) and 2 for COMM_ECHAM (blue).

For the ensemble DA, the parallelization of AWI-CM is modified. Generally, the introduction of the ensemble adds one additional level of parallelization to a model, which allows us to concurrently compute the ensemble of model integrations, i.e. several concurrent model tasks. In AWI-CM augmented by the calls to PDAF, the routine *init_parallel_pdaf* modifies the parallelization. Namely MPI_COMM_WORLD is split into a group of communicators for the coupled model tasks (COMM_CPLMOD), as exemplified for an ensemble of 4 model tasks in Fig. 3(b) indicated by the different color shading.

Subsequently, OASIS-MCT splits each communicator COMM_CPLMOD into a pair COMM_ECHAM and COMM_FESOM (third line in Fig. 3(b)). To be able to split COMM_CPLMOD, the source code of OASIS-MCT needs to be modified replacing MPI_COMM_WORLD by COMM_CPLMOD, because OASIS-MCT uses MPI_COMM_WORLD as the basis for the communicator splitting (see also Kurtz et al., 2016, for the required modifications). With this configuration of the communicators, AWI-CM is able to integrate an ensemble of model states by computing all model tasks concurrently.

Two more communicators are defined in *init_parallel_pdaf* for the analysis step in PDAF. Here, a configuration is used that computes the filter analysis step on the first coupled model task using the same domain-decomposition as the coupled model. Because the ESTKF (as any other ensemble Kalman filter) computes a combination of all ensemble members individually for each model grid point or for single vertical columns (Eq. 1), the ensemble information from all ensemble members is collected on the processes of the first model task, keeping the domain decomposition. For collecting the ensemble information,

the communicator COMM_COUPLE groups all processes that compute the same sub-domain in the coupled model. Thus, all processes that have the same rank index in e.g. COMM_FESOM are grouped in one communicator as shown in line 4 of Fig. 3(b). Finally, the communicator COMM_FILTER (line 5 of Fig. 3(b)) is defined, which contains all processes of the first model task. Note that compared to the single-compartment case discussed in Nerger et al. (2005) and Nerger and Hiller (2013), the major change is that each model task is split into the communicators COMM_FESOM and COMM_ECHAM,

which are, however, only used for the model integration. In addition, COMM_FILTER includes the processes of both model compartments of the first model task.

This configuration is used to perform strongly-coupled DA, because it allows the communication between processes of ECHAM with processes of FESOM. In a weakly-coupled application of DA, COMM_FILTER is initialized so that two separate communicators are created, one for all sub-domains of FESOM and another one for all sub-domains of ECHAM as shown in

Fig. 3(c). In practice one can achieve this by using the already defined communicators COMM_FESOM and COMM_ECHAM of model task 1. Because these two communicators are initialized after executing *init_parallel_pdaf*, one has to overwrite COMM_FILTER afterwards in, e.g., *init_PDAF*. With this configuration the assimilation can be performed independently for both compartments.

### 3.4 Call-back routines for handling of model fields and observations

The call-back routines are called by PDAF to perform operations that are specific for the model or the observations. The operations performed in each routine are rather elementary to keep the complexity of the routines low. There are four different types of routines, which are displayed in Fig. 4:

- *interfacing model fields and state vector (cyan)*: There are two routines called before and after the analysis step. The first routine writes model fields into the state vector of PDAF, while the second initializes model fields from the state vector.
These routines are executed by all processes that participate in the model integrations and each routine acts on its process sub-domain. For the coupled model, there are different routines for FESOM and ECHAM.

- *observation handling (orange)*: These routines perform operations related to the observations. For example, a routine provides PDAF with the number of observations, which is obtained by reading the available observations and counting them. This routine allows PDAF to allocate arrays for the observed ensemble. Another routine is the implementation
of the observation operator. Here, the routine is provided with a state vector $\mathbf{x}$ from the ensemble and has to return the observed state vector, i.e. $\mathbf{H}(\mathbf{x})$. For the coupled model, the routines are distinct for FESOM and ECHAM as, e.g., the observation operator for an oceanic observation can only be applied in FESOM. For strongly coupled DA, the observation operator routine would also contain parallel communication that acts across the compartments. Thus, after obtaining the observations in a compartment, a cross-compartment observation vector is initialized using MPI communication.

- *localization (yellow)*: The localized analysis described in Sec. 2.1.1 requires several operations, which are provided by call-back routines. For example, a call-back routine needs to determine the dimension of a local state vector. For a single grid point this would be the number of variables stored at this grid point. For a vertical column of the model grid, this would be the number of 3-dimensional model fields times the number of model layers plus the number of 2-dimensional model fields (like sea surface height or sea ice variables in FESOM). Then, after PDAF allocates the local state ensemble,
a call-back routine is used to fill the local states from the full domain-decomposed state vector (likewise, there is a routine that writes a local state vector after the local analysis correction into the full state vector). In addition, there is a routine that determines the number of observations within the influence radius around the vertical column and a routine to fill this local observation vector from a full observation vector.

- *pre- and post-processing (blue)*: To give the user access to the ensemble before and after the analysis step, there is a
375 pre/post-processing routine. Here, one typically computes the ensemble mean and writes it into a file. Further, one could implement consistency checks, e.g. whether concentration variables have to be positive, and can perform a correction to the state variables if this is not fulfilled.

## 4 Parallel performance of the coupled data assimilation system

### 4.1 Scalability

To assess the parallel performance of the assimilation system described above, AWI-CM is run here in the same global configuration as described by Sidorenko et al. (2015). The atmosphere uses a horizontal spectral resolution T63 (about 180 km) with 47 layers. The ocean model uses an unstructured triangular grid with 46 vertical layers. The horizontal resolution varies between 160 km in the open ocean, with a refinement to about 45 km in the equatorial region and close to the Antarctic continent, and 30 km north of $50^o$ N. The models are run with a time step size of 450 seconds for ECHAM and 900 seconds for FESOM. The coupling by OASIS-MCT is performed hourly.

In the initial implementation AWI-CM-PDAF 1.0, the assimilation update is only performed as weakly coupled DA in the ocean compartment. The state vector for the assimilation is composed of the 2-dimensional sea surface height, and the 3-dimensional model fields temperature, salinity and the three velocity components. The DA is started on January 1st, 2016 and satellite observations of the sea surface temperature obtained from the European Copernicus initiative (data set SST_GLO_SST_L3S_NRT_OBSERVATIONS_010_010 available at https://marine.copernicus.eu), interpolated to the model grid, are assimilated daily. The assimilation is multivariate so that the SST observations influence the full oceanic model state vector through the ensemble estimated cross-covariances that are used in the ESTKF. The initial ensemble was generated using second-order exact sampling (Pham et al., 1998) from the model variability of snapshots at each 5th day over one year. the ensemble mean was set to a model state for January 1, 2016 from a historical (climate) run of AWI-CM. No inflation was required in this experiment, i.e. a forgetting factor $\rho = 1.0$ (see Eq. 4) was used. Even though, we only perform weakly coupled DA here, we expect that the compute performance would be similar in case of strongly coupled DA, as is explained in Sec. 6.

For a fixed ensemble size but varying number of processes for ECHAM and FESOM, the scalability of the program is determined by the scalability of the models (see, e.g., Nerger and Hiller, 2013). To access the scalability of the assimilation system for varying ensemble size, experiments over 10 days were conducted with varying ensemble sizes between $N_e = 2$ and $N_e = 46$. The length of these experiments is chosen to be long enough so that the execution time is representative to assess the scalability. However, the assimilation effect will be rather small for these 10 analysis steps. The number of processes for each model task was kept constant at 72 processes for ECHAM and 192 processes for the more costly FESOM. The experiments were conducted on the Cray XC40 system 'Konrad' of the North-German Supercomputer Alliance (HLRN).

Fig. 5 shows the execution times per model day for different parts of the assimilation program. Shown are the times for 24-hour forecast phases including the time to collect and distribute the ensemble (DA coupling within the communicator COMM_COUPLE) for the analysis step. Also shown are the times for the analysis step (green), the execution of the pre-/post-step operations (red), and the DA coupling time (blue). The crosses show the time for each model task and separately for the atmosphere and ocean, thus there are $2N_e$ black and blue crosses for each ensemble size. The blue and black lines show the maximum execution times. The overall execution time is dominated by the time to compute the forecasts. The combined time for the analysis and the pre/post step operations is only between 4 and 7% of the forecast time. For a given ensemble size, the black crosses show that the execution times for the forecast on the different model tasks vary. In the experiments, the longest

forecast time was up to 16% larger than the shortest time, which occurred for $N_e = 24$. This variability is partly caused by the time for DA coupling (see discussion below), but also by the fact that the semi-implicit time stepping of FESOM leads to varying execution times. Further influence have the parallel communication within each compartment at each time step and the communication for the model coupling by OASIS3-MCT that is performed at each model hour. The execution time for these operations depends on how the overall program is distributed over the computer. As the computer is also used by other applications, it is likely that the application is widely spread over the computer so that even different compute racks are used. This can even lead to the situation that the processors for a single coupled model task of ECHAM and FESOM, but also a single model instance of ECHAM or FESOM, are not placed close to each other. If the processors are distant, e.g. in different racks, the communication over the network will be slower than for a compact placement of the processors. To this end, also the execution time will vary when an experiment for the same ensemble size is repeated. Nonetheless, repeated experiments showed that the timings in Fig. 5 are representative. Likewise experiments in the new supercomputer system 'Lise' of the HLRN showed similar timings, though the forecast time was reduced to about 27 seconds per day compared to about 35 seconds shown in Fig. 5.

The variation of the forecast time when the ensemble size is changed is mainly caused by the varying time for the DA coupling. When the time for the DA coupling is subtracted from the forecast time, the variability is much reduced as the black dashed line shows. The variability in dependence on the ensemble size is better visible when the execution time is normalized relative to the time for $N_e = 2$ as is displayed in Fig. 6. The forecast time including DA coupling fluctuates and increases by up to 8% for the largest ensemble with $N_e = 46$ (black line). In contrast, the forecast time without DA coupling only increases by about 3.5% (black dashed line). The time for the DA coupling (blue line) varies by a factor of 2.5. This large variation is due to the fact that here the communication happens in the communicators COMM_COUPLE, which are spread much wider over the computer than the communicators for each coupled model task (COMM_CPLMOD) as is visible in Fig. 3. However, even though the number of ensemble states to be gathered and scattered in the communication for the DA coupling varies between 2 and 46, there is no obvious systematic increase in the execution time. In particular, for $N_e = 40$ the execution time is almost identical to that of $N_e = 2$.

Further variation in dependence on the ensemble size is visible for the pre-/post-step operations (red line). This variation is mainly due to the operations for writing the ensemble mean state into a file. In contrast, the analysis step shows a systematic time increase. The time for computing the analysis for $N_e = 46$ is about seven times as long as for $N_e = 2$. This is expected from the computational complexity of the LESTKF algorithm (see Vetra-Carvalho et al., 2018). However, also the LESTKF performs MPI communication for gathering the observational information from different process domains. When repeating experiments with the same ensemble size we found a variation of the execution time for the analysis step of up to 10%.

## 4.2 Performance tuning

To obtain the scalability discussed above important optimization steps have been performed. First, it is important that each coupled model instance is, as far as possible, placed compactly in the computer. Second, one has to carefully consider the disk operations performed by the ensemble of coupled model tasks.

For the first aspect, one has to adapt the run script. The coupled model is usually started with a command line like

$mpirun - np \ N_O \ fesom.x \ : \ -np \ N_A \ echam.x$

(or any other suitable starter for an MPI-parallel program) such that FESOM and ECHAM are run using $N_O$ and $N_A$ processes, respectively. For the DA one could simply change this by replacing $N_O$ by $N_e \times N_O$ and $N_A$ by $N_e \times N_A$ to provide enough
processes to run the ensemble. This is analogous to the approach used when running a single-compartment model. However, changing the command line in this way will first place all MPI tasks for the FESOM ensemble in the computer followed by all MPI tasks for the ECHAM ensemble. Accordingly, each ocean model will be placed distant from the atmospheric model to which it is coupled. Using this execution approach, the time for the forecasts discussed above increased by a factor of four, when the ensemble size was increased from 2 to 46. For a more efficient execution, one has to ensure that the ocean-atmosphere
pairs are placed close to each other. This is achieved with a command line like

$mpirun - np \ N_O \ fesom.x \ : \ -np \ N_A \ echam.x \ : \ -np \ N_O \ fesom.x \ : \ -np \ N_A \ echam.x \ \ldots$

which contains as many FESOM-ECHAM pairs as there are ensemble members. With this approach, the time increase of the forecast was reduced to about 40% for the increase from $N_e = 2$ to $N_e = 46$.

For the second issue regarding disk operations, one has to take into account that the direct outputs written by each coupled
ensemble task are usually not relevant because the assimilation focuses on the ensemble mean state. To this end, one generally wants to deactivate the outputs written by the individual models and replace them by outputs written by the pre-/post-step routine called by PDAF. If the model does not allow to fully switch off the file output, it usually helps to set the output interval of a model to a high value (e.g. a year for a year-long assimilation experiments). However, in case of AWI-CM this strategy still resulted in conflicts of the input/output operations so that the models from the different ensemble tasks tried to write into the
same files, which serialized these operations and increased the execution time. To avoid these conflicts it helped to distribute the execution of the different ensemble tasks to different directories, e.g.

$mpirun - np \ N_O \ 01/fesom.x \ : \ -np \ N_A \ 01/echam.x \ : \ -np \ 02/N_O \ fesom.x \ : \ -np \ N_A \ 02/echam.x \ \ldots$

combined with a prior operation in the run script to generate the directories and distribute the model executables and input files. This distribution avoids that two model tasks writes into the same file and improves the performance of the ensemble
DA application. In this configuration, the performance results of Sec. 4.1 were obtained. Another benefit of separate execution directories is that ensemble restarts can be easily realized. Given that each model task write its own restart files in a separate directory, a model restart is possible from these files without any adaptions to the model code. Note, that the approach of separate directories is also possible for the ensemble DA in case of a single (uncoupled) model like a FESOM-only simulation using atmospheric forcing data as e.g. applied by Androsov et al. (2019).

# 5   Application Example

Applications based on the AWI-CM-PDAF 1.0 code are Mu et al. (2020), where the focus is on the effect on sea ice, and Tang et al. (2020), who discuss the reaction of the atmosphere on assimilating ocean observations.

Here, we demonstrate the functionality of the data assimilation system in an experiment assimilating SST data over the year 2016. An ensemble of 46 states is used, which is the maximum size, used in the scalability experiment discussed above. The assimilation is performed with a localization radius of 500 km using the regulated localization function by Nerger et al. (2012a). The same SST observations as in Sec. 4.1 are assimilated, which are treated as in Tang et al. (2020). The resolution of the observations is $0.1^o$ and hence higher than the resolution of the model in most regions. Since the model grid is unstructured with varying resolution, super-observations are generated by averaging onto the model grid. The observation error standard deviation for the assimilation was set to $0.8^oC$ and observations whose difference from the ensemble mean is more than two standard deviations are excluded from the assimilation. This approach excludes about 22% of the observations at the initial first analysis step. The number of excluded observations shrinks during the course of the assimilation and after one month less than 5% of the days observations are excluded. The assimilation further excludes observations at grid points for which the model contains sea ice because of the mis-match of the satellite data representing ice-free conditions, while ice is present on modeled ocean surface. Two experiments are performed: The experiment FREE runs the ensemble without assimilating observations while the experiment DA-SST assimilates the SST data.

Figure 7 shows root mean square error (RMSE) of the SST in the analysis step with respect to the assimilated observations over time. Given that the SST observations are assimilated it is a necessary condition for the DA to reduce the deviation from these observations. At the initial analysis time (i.e. after 24 hours), the RMSE is about $1.2^oC$. In the free run, the RMSE increases first to about $1.4^oC$ and reaches nearly $1.6^oC$ a the end of the year. The assimilation in DA-SST strongly reduces the RMSE during the first two months. During this initial transient phase, the RMSE is reduced to about $0.45^oC$. Afterwards, the RMSE remains nearly constant, which is a typical behavior. On average over the year 2016, the RMSE in the experiment DA-SST is $0.51^oC$, while it is $1.38^oC$ for the free run.

To validate the assimilation with independent observations, temperature and salinity profiles from the EN4 data set (EN4.2.1) of the UK MetOffice (Good et al., 2013) are used. This collection of in situ data contains about 1000 to 2000 profiles per day at depths between the surface and 5000 m depth. Figure 8 shows the RMSE of the experiment DA-SST relative to the RMSE for the free run. Hence values below one indicate improvements. For the temperature a gradual improvement is visible during the first 100 days. The error reduction reach about 40% during the year. On average, the RMSE is reduced by 14% from $1.85^oC$ to $1.40^oC$. The variations in the RMSE, e.g. the elevated values around day 250 are due to the varying coverage and location of the profiles in the EN4 data set. For the salinity the effect of the DA is lower. While the RMSE of the salinity first increases during the first month, it is reduced from day 60, but until day 140 it is sometimes larger than at the initial time. Partly the RMSE is reduced by up to 23% at day 144. On average over the full year of the experiment, the RMSE of salinity is reduced by 5.6%. This smaller effect on the salinity is expected because there are no strong correlations between the SST and the salinity at different depths. The improvements of the model fields by the DA of SST is mainly located in the upper 200m of the ocean. For the temperature the RMSE is reduced by 15.2% in the upper 200m, but only 3.0% below 200m. This is also an expected effect because the correlations between SST and subsurface temperature are largest in the mixed layer of the ocean.

## 6 Discussion

The good scalability of the assimilation system allows to perform the assimilation experiment of Sec. 5 over one full year with daily assimilation in slightly less than 4 hours, corresponding to about 53,000 core-hours. As such the system is significantly faster than the coupled ensemble DA application by Karspeck et al. (2018), who reported to complete one year in 3 to 6 weeks with an ensemble of 30 states and about one million core-hours per simulation year. However, both systems are not directly comparable. Karspeck et al. (2018) used atmospheric and ocean models with 1° resolution. Thus the atmosphere had a higher resolution than used here, while the ocean resolution was comparable to the coarse FESOM resolution in the open ocean, which was then regionally refined. Given that both model compartments in AWI-CM scale to larger processor numbers than we used for the DA experiment, we expect that the DA into AWI-CM with ECHAM at a resolution of T127 (i.e. about 1°) could be run at a similar execution time as for T63 given that a higher number of processors would be used. Further Karspeck et al. (2018) applied the DA also in the atmosphere, while here only oceanic data was assimilated. Given that the atmospheric analysis step would typically be applied after each 6th hour, the time for the DA coupling and the analysis steps would increase. However, we don't expect that a single atmospheric analysis step would require significantly more time than the ocean DA so that due to the parallelization the overall run time should not increase by more than 10-20%. Further, we expect a similar scalability in case of strongly coupled DA. The major change for strongly coupled DA is to communicate the observations in between the compartments as mentioned above. This communication will only be small part of the analysis time.

Important for the online-coupled assimilation system is that there is obviously no significant time required for re-distributing the model field (i.e. the time for the DA coupling discussed in Sec. 4.1). Furthermore there is no transpose of the ensemble array to be performed, which was reported to be costly by Karspeck et al. (2018). Here, the implementation of the analysis step uses the same domain-decomposition as the models and hence only the full ensemble for each process sub-domain has to collected by the DA coupling. Thus, only groups of up to 46 processes communicate with each other in this step.

The online-coupled assimilation system avoids any need for frequent model restarts. Actually, the initial model startup of AWI-CM took about 95 seconds and the finalization of the model with writing restart files tool another 15 seconds. Thus, these operations take about 3.3 times longer than integrating the coupled model for one day. If the DA would be performed in a separate program coupled to AWI-CM through files, these operations would be required each model day. In addition, the assimilation program would also need to read these restart files and write new restart files after the analysis step. Assuming that these observations take about 15 seconds, like the finalization of the coupled model, the execution time would increase by a factor of 4 for offline-coupled DA compared to online-coupled DA.

The code structure using interface routines inserted into the model code and case-specific call-back routines makes the assimilation framework highly flexible. Further, the abstraction in the analysis step, which uses only state and observation vectors without accounting for the physical fields allows one to separate the development of advanced DA algorithms from the development of the model. Thus, a separation of concerns is ensured, which is mandated for efficient development of complex model codes and their adaptions to modern computers (Lawrence et al., 2018). The separation allows that, as soon as a new DA method is implemented, all users with their variety of models can use this method by updating the PDAF library. To ensure

compatibility of different versions of the library, the interfaces to the PDAF routines are kept unchanged. However for a new filter additional call-back routines might be required, e.g. a routine to compute the likelihood of an ensemble according to the available observations in case of the nonlinear ensemble transform filter (NETF, Tödter and Ahrens, 2015) or a particle filter. The abstraction in the analysis step and the model-agnostic code structure also allow to apply the assimilation framework independent of the specific research domain. E.g. applications of PDAF with a geodynamo model (Fournier et al., 2013),

hydrological applications (Kurtz et al., 2016), ice shield modeling (Gillet-Chaulet, 2020), and volcanic ash clouds (Pardini et al., 2020) have been published.

The example here uses a parallelization in which the analysis step is computed using the first model task and the same domain decomposition as the model. Other parallel configurations are possible. E.g., one could compute the analysis step not only using the processes of model task 1, but for processes of several or all model tasks. This could be done by either using a finer domain-

555 decomposition than in the model integrations, or by e.g. distributing different model fields onto the processes. These alternative parallelization strategies are, however, more complex to implement and hence not the default in PDAF. A further alternative, which is already supported by PDAF, is to dedicate a set of processes for the analysis step. In this case, the DA coupling would communicate all ensemble members to these separate processes. However, these processes would idle during the forecast phase. To this end, separating the processes for the analysis step would mainly be a choice if the available memory on the

560 first model task is not sufficient to execute the analysis step. Also in this case, the distribution of the analysis step over several processors would reduce the required memory. For the parallel configuration of AWI-CM-PDAF in Fig. 3, a particular order of the processes is assumed. This order originates from the startup procedure of MPI and is determined by the command line which start the program. Thus, for other models one might need a different setup, which can usually be obtained by only modifying the routine *init_parallel_pdaf*. Further, the default version of this routine splits the communicator MPI_COMM_WORLD.

However, for other models a different suitable communicator might be split if not all processes participate in the time stepping. This can be the case when, e.g., an OI-server is used that reserves processes exclusively for the file operations. To provide flexibility to adapt to such requirements, the routine *init_parallel_pdaf* is compiled with the model and is not part of the core routines of the PDAF library.

While the fully-parallel execution of the assimilation program is very efficient, it is limited by the overall job size allowed

on the computer. The maximum ensemble size was here limited by the batch job size of the used computer. The model used in the example here can scale even further than e.g. the 192 processes used for FESOM and 72 processes for ECHAM. Thus, using the same computer, one could run a larger ensemble with less processes per model and accordingly a larger run time, or a smaller ensemble with less run time. The number of processes should be set so that the requirements on the ensemble size for a successful assimilation can be fulfilled. Nonetheless, the ensemble DA is computationally demanding and for larger

applications, one might need to obtain a compute allocation at larger computing sites, like national compute centers.

## 7    Conclusions

This study discussed the parallel data assimilation framework (PDAF) and its use to create a coupled data assimilation program by augmenting the code of a coupled model and using in-memory data transfers between the model and the data assimilation software. The implementation strategy was exemplified for the coupled ocean-atmosphere model AWI-CM for which two separate programs for the ocean and atmosphere where augmented. However, the strategy can be easily used for other model systems consisting of a single or multiple executables.

The implementation of a DA system based on PDAF consist in augmenting the model codes with calls to routines of the assimilation framework. These routines modify the parallelization of the model system, so that it becomes an ensemble model. Further, the ensemble is initialized and the analysis step of the data assimilation can be executed at any time without restarting the model. Operations to transfer between model fields and the abstract state vector of the assimilation, and the observation handling are performed in case-specific routines. These routines are executed as call-back routines and can be implemented like routines of the numerical model, which should simplify their implementation.

Numerical experiments with daily assimilation of sea surface temperature observations into the AWI-CM showed an excellent scalability when the ensemble size is increased. This resulted in an overhead which was, depending on the ensemble size, only up to 15% in computing time compared to the model without assimilation functionality. The execution time of the coupled ensemble data assimilation program was dominated by the time to compute the ensemble integrations in between the time instances at which the observations are assimilated. This excellent scalability resulted from avoiding disk operations by keeping the ensemble information in memory and exchanging it through parallel communication during the run time of the program. Care has to be taken that in the coupled model the pairs of atmosphere and ocean model compartments are placed close to each other in the computer, which can be achieved by specifying these pairs in the command starting the parallel program. The time to collect this ensemble information before the analysis step and to distribute it afterwards showed significant variations from run to run. These variations are due to the fact that the large compute application is widely spread over processors of the computer. Anyway, no systematic time increase was observed when the ensemble size was increased and the time was only up to about 6% of the time required for the forecasting. Distributing the different models over separate directories improved the scalability because it avoided possible conflicts the in file handling which can be serialized by the operating system of the computer.

PDAF provides a model-agnostic framework for the efficient data assimilation system as well as filter and smoother algorithms. As such it provides the capacity to ensure a separation of concerns between the developments in the model, observations, and the assimilation algorithms. Functionality to interface between the model, which operates on physical fields, and the assimilation code, which only work on abstract state vectors, has to be provided in a case-specific manner by the users based on code templates. This also holds for the observation handling. While there are typical observational data sets for the different Earth system compartments, the observation operator links the observations with the model fields on the model grid. Thus, the observation operator has to be implemented taking into account the specific character of the model grid like the unstructured structure of FESOM's grid.

The current implementation of AWI-CM-PDAF only contains the assimilation into the ocean component, while the assimilation into the atmosphere is technically prepared. First studies (Mu et al., 2020; Tang et al., 2020) base on this implementation. In future work we plan to add the assimilation of atmospheric observations, and to complete the implementation of strongly-coupled data assimilation, which requires the exchange of observations in between the ocean and atmosphere.

*Code availability.* The model-binding for AWI-CM-PDAF 1.0 used in this study is archived at Zenodo (Nerger et al., 2019a) The PDAF code (version 1.14 was used here), as well as a full code documentation and a usage tutorial are available at http://pdaf.awi.de. The source code of the coupled AWI-CM model (revision 550 was used) is available from from the SVN repository at https://swrepo1.awi.de/svn/awi-cm/trunk@550 (last access: May 2020) and can be downloaded using SVN. The ECHAM6 source code is maintained by the Max Planck Institute for Meteorology and is freely available to the public (http://www.mpimet.mpg.de/en/science/models/mpi-esm/echam/, Max Planck Institute for Meteorology, 2019a). External access to the ECHAM6 model is provided through their licensing procedure (http://www.mpimet.mpg.de/en/scien Only after registering for using ECHAM6, access to AWI-CM can be granted. The OASIS3-MCT coupler is available for download at https://portal.enes.org/oasis (ENES Portal, 2011).

*Data availability.* The experiments have been performed using the LR mesh of FESOM. For the availability of this configuration, mesh, and input files see Rackow et al. (2019). The output files containing the timing information, the outputs from the 1-year experiments, and plotting scripts are available at Zenodo (Nerger et al., 2019b)

*Author contributions.* The main body of the manuscript was written by LN, with inputs from the co-authors. LN also leads the development of PDAF and developed the assimilation coupling strategy. QT implemented PDAF with AWI-CM and performed the timing experiments. Both QT and LM worked on optimizing the compute performance of the implementation of PDAF with AWI-CM. LM further developed the restarting functionality based on separate run directories.

*Competing interests.* The authors declare that they have no conflict of interest.

*Acknowledgements.* We thank the North-German Supercomputing Alliance (HLRN) for providing compute resources (project hbk00064). This work is funded by the project ESM - Advanced Earth System Modeling Capacity of the German Helmholtz-Association. We are grateful to Dmitry Sidorenko for support in the setup of the AWI-CM model. We like to thank the two anonymous reviewers and the Editor for their comments that helped to improve the manuscript.

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

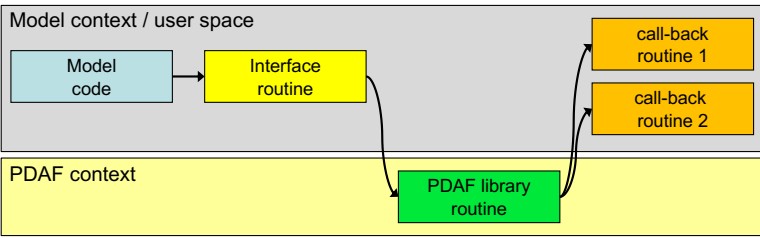

**Figure 1.** Call-structure of PDAF. Calls to interface routines (yellow) are inserted to the model code (blue). The interface routines define parameters for PDAF and call PDAF library routines (green). These library routines call user-provided call-back routines. The model code, interface, and call-back routines operate in the model context and can hence exchange information indirectly, e.g. through Fortran modules. Likewise, the PDAF library routines share variables.

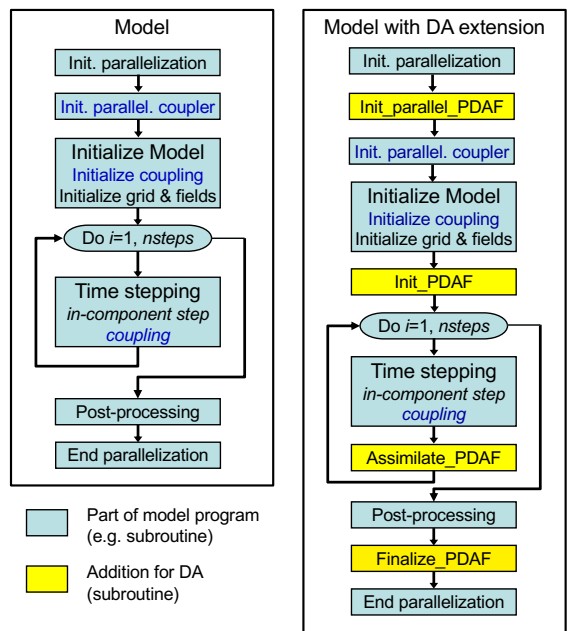

**Figure 2.** General program flow: (left) abstract original program without data assimilation; (right) program augmented for data assimilation. The blue color marks coupling routines whose parallelization needed to be adapted for the data assimilation. Each of the two coupled compartment models were augmented in this way.

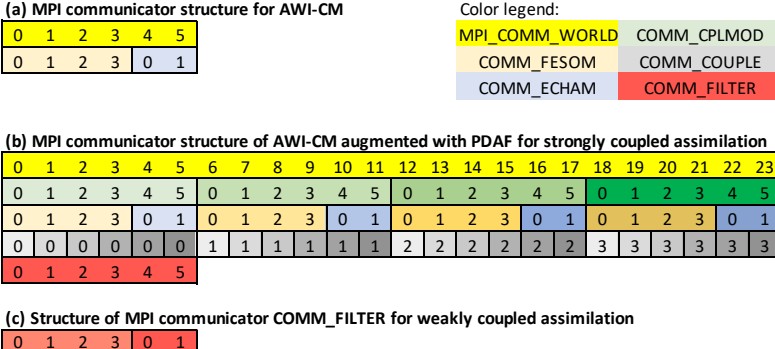

**Figure 3.** Example configuration of MPI communicators: (a) AWI-CM, (b) AWI-CM with PDAF-extension for ensemble data assimilation. The colors and lines mark processes that are grouped as a communicator. Different shades of the same color mark the same communicator type (e.g. four orange communicators COMM_FESOM). For COMM_COUPLE each communicator is spread over the model tasks. The numbers mark the rank index of a process in a communicator.

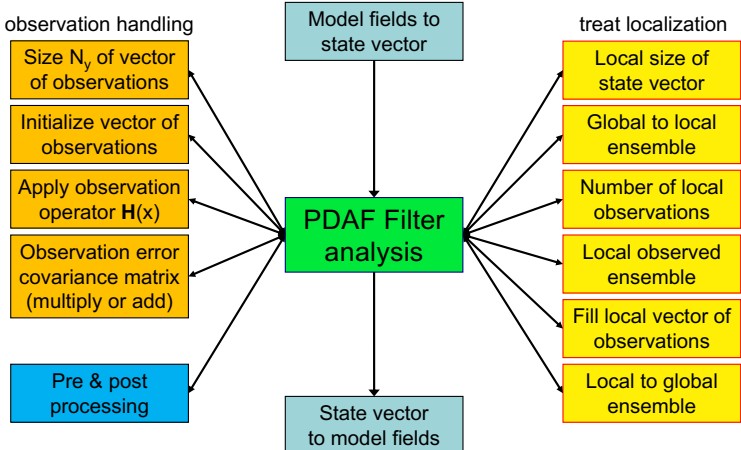

**Figure 4.** PDAF filter analysis step and related call-back routines provided by the user. there are four types of routines: transfers between model fields and state vector (cyan), observation handling (orange), treatment of localization (yellow), and pre/post-processing (blue).

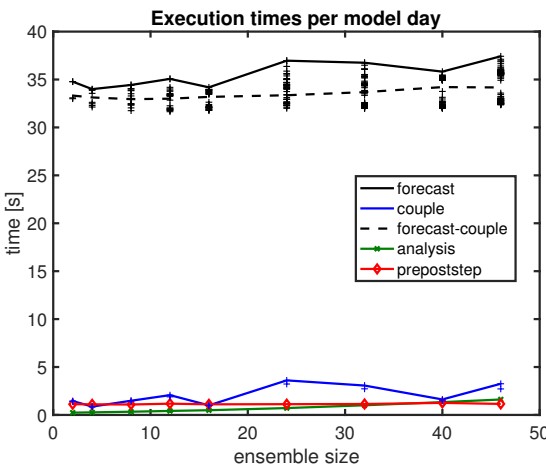

**Figure 5.** Execution times per model day for varying ensemble sizes for different parts of the assimilation program. The dominating forecast time includes the 'coupling' time which results in the time variations.

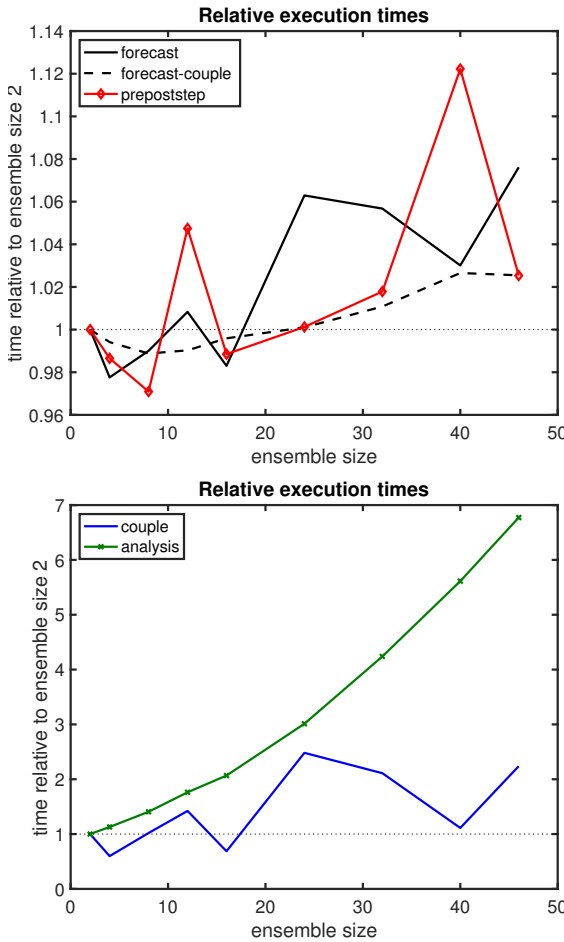

**Figure 6.** Execution times relative to ensemble size 2 for different parts of the assimilation program as a function of the ensemble size. The fluctuation is the time is caused by parallel communication and file operations. The analysis step shows a systematic time increase, while the time for DA-coupling varies strongly.

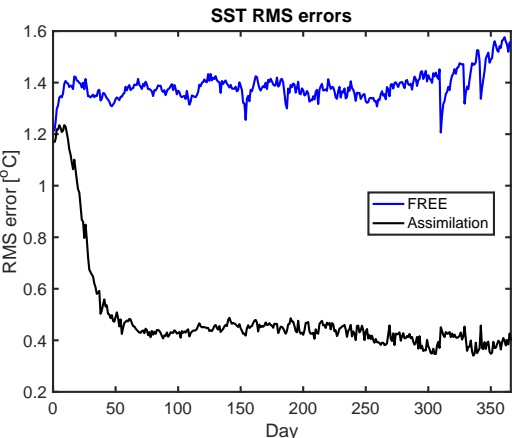

**Figure 7.** Root mean square errors of the estimated SST with regard to the assimilated SST observations. Shown are the free run (blue) and the SST assimilation experiment (black). During the spin-up period of the DA, the RMS erros are strongly reduced.

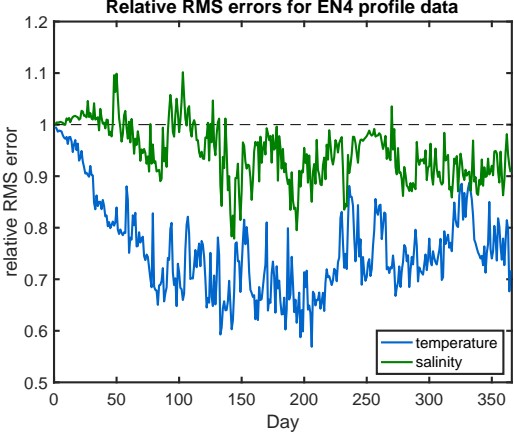

**Figure 8.** Root mean square errors (RMSEs) of the assimilation experiment relative to the free run computed with regard to the in situ EN4 profile observations. Shown are the relative RMSEs for temperature (blue) and salinity (green). Temperature is more strongly improved than salinity.