# Peer review of "Efficient ensemble data assimilation for coupled models with the Parallel Data Assimilation Framework: Example of AWI-CM (AWI-CM-PDAF 1.0)"

_Geoscientific Model Development, 2019_

## Short Comment (SC1) · 7 Aug 2019

Dear authors,

in my role as Executive editor of GMD, I would like to bring to your attention our Editorial version 1.2:

https://www.geosci-model-dev.net/12/2215/2019/

This highlights some requirements of papers published in GMD, which is also available on the GMD website in the 'Manuscript Types' section: http://www.geoscientific-model-development.net/submission/manuscript_types.html

In particular, please note that for your paper, the following requirement has not been met in the Discussions paper:

- "The main paper must give the model name and version number (or other unique identifier) in the title."

Please provide the version number for PDAF in the title of your revised manuscript.

Additionally, please note, that GMD is encouraging authors to provide a persistent access to the exact version of the source code used for the model version presented in the paper. As explained in https://www.geoscientific-model-development.net/about/manuscript_types.html the preferred reference to this release is through the use of a DOI which then can be cited in the paper. For projects in GitHub a DOI for a released code version can easily be created using Zenodo, see https://guides.github.com/activities/citable-code/ for details.

Finally note, that according to our new Editorial (v1.2) all data and analysis / plotting scripts should be made available.

Yours,

Astrid Kerkweg

---

## Referee Comment (RC1) · Anonymous Referee #1 · 16 Oct 2019

The paper describes the implementation of the software tool PDAF to a coupled ocean-atmosphere model. It discusses essentially the general structure of the PDAF software and how the coupling can be realized on a distributed computing architecture with MPI. While this is interesting, my main issues with this manuscript are the following 4 points:

1. No actual results of the assimilation system are presented. Only the execution time for different settings. It is unclear to me what the role of a reviewer can be in this case. I rather think that the paper should also include the results of such model (see also the following point).

2. The manuscript mentions different approaches to implement the assimilation in

a coupled system: in a combined state vector spanning the atmospheric and ocean model or separately. The question about which approach is better is still open and it should not be too difficult to the authors to check both approaches. This would help also to address the previous point and add substantially to the scientific value of this paper.

3. The different time scales of ocean and atmosphere are not discussed and the assimilation is done only in the ocean. To really appreciate the effectiveness of the coupling, data should be assimilated in both the atmosphere and the ocean and the question regarding the assimilation frequency should be addressed. As usual, the models should be validated against independent observations.

4. There is too much overlap between this manuscript and previous manuscripts by the same author concerning the description of PDAF (in particular the memory coupling, general API structure). I think the author should focus this paper on the coupling aspect and just reference to elements already published before.

I therefore recommend major revision before this article is published in GMD.

Minor comments:

line 46: tranDAsfers → transfer

page 6: MPI Communicators: is this discussion not too technical?

Section 5: How the system scales for a fixed ensemble size?

Figure 6: the label mentions relative execution times, but the unit on the axis is [s].

———————————

---

## Short Comment (SC2) · 19 Oct 2019

Dear authors,

Thanks a lot for making this article of the latest improvement of PDAF online on GMD for discussion.

I am Li Liu from Tsinghua University, China, leading the development of C-Coupler that is a Chinese coupler family for Earth system modelling. I am very interested in the software framework for ensemble data assimilation, even leading a research in this topic. So I am very interested in your PDAF work, and have learned a lot from its

documentations, source codes, and this article.

After reading this article, I have the following concerns:

1. After downloading and then reading the latest available code version of PDAF from your website, I guess that it does not fully include the implementation for this article.

2. Figure 3 of this article and the source code of PDAF may indicate that the certain order of processes in the MPI_COMM_WORLD among ensemble members of the coupled model as well as its component models is required, and different members of the same component model must have the same number of processes. For example, the IDs of processes of atm_member1, ocn_member1, atm_member2, ocn_member2, atm_member3 and ocn_member3, and the processes not involved in ensemble data assimilation, must be in an ascending/descending order. Is there any restriction about the processes not involved in ensemble data assimilation?

3. Regarding P9L261~270, it seems unclear how to split the communicator for a set of processes exclusive from ensemble data assimilation when splitting MPI_COMM_WORLD into a group of COMM_CPLMOD. Is there any new modification in the APIs or input files of PDAF for this functionality?

4. Regarding the weakly-coupled application mentioned in P10L285~L288, it seems unclear how to generate separate COMM_FILTER for ECHAM and FESOM? Regarding this functionality, I guess that PDAF should know all component models of the coupled model and the number of processes of each component model, and know that weak coupling but not strong coupling is used. It may be interesting to know how PDAF is extended for this kind of input.

5. Regarding Figure 2, it is still unclear of the code flowchart of different component models in weak coupling. For example, given that only ECHAM is involved in data assimilation but FESOM is not, it is unclear whether only ECHAM calls init_PDAF and Assimilate_PDAF, or FESOM has to call these two APIs cooperatively?

6. PDAF requires the filter to use the same parallel decomposition with the model. Is it possible to introduce challenges when integrating an existing filter that already has its own parallel decomposition that may be different from the model. For example, a land surface model generally uses a round-robin parallel decomposition for load balance in parallelization, which may be not suitable for a filter or will introduce new code development or lower efficiency to the filter.

I really hope these concerns can be further discussed in the final version of this article. I am sorry if some of them are incorrect or even wrong.

Best regards,

Li

---

## Referee Comment (RC2) · Anonymous Referee #2 · 21 Oct 2019

The manuscript describes the application of the Parallel Data Assimilation Framework (PDAF) for coupled data assimilation, with a strong focus on strongly-coupled data assimilation (DA). An example implementation with a coupled atmosphere-ocean model is described in detail and the differences to a previous similar application of PDAF as well as to a similar application of the Data Assimilation Research Testbed are explained and discussed.

While the presented MPI-based implementation for strongly-coupled data assimilation with PDAF is a logical extension of PDAF's approach for single-component models, it merits publication as a novel and highly relevant approach in the coupled case. This

is well demonstrated by the comparison to and discussion of the implementations in Kurtz et al. 2016 and Karspeck et al. 2018.

However, the presented example of data assimilation for the coupled atmosphere-ocean model AWI-CM seems to fall short of demonstrating strongly-coupled data assimilation. Lines 322 to 330 describe a weakly-coupled assimilation system with coupled forecasts but observations of and assimilation in the ocean component only. The text explicitly states that "the assimilation update is only performed in the ocean compartment" which is confusing after sections 2.2 and 3.3 describe how the model states of ocean and atmosphere components are joined into a single state vector and how the model codes are extended to realize this technically. Presumably this experiment could have been realized with less code modifications than mentioned in the text. While even this setup with ocean-only assimilation into a coupled model demonstrates progress over data assimilation into a single-component model, the current presentation is unfortunate.

I suggest that either the use of the presented example is well justified in the text and its relation to the previous sections and strongly-coupled DA is explained or that the example is extended to a strongly-coupled DA experiment. As it appears that large parts of the discussion and conclusion would still apply to a truly strongly-coupled data assimilation experiment, I would encourage the authors to aim for this way forward.

Other minor points/typos:

line 46: transfers instead of tranDAsfers

line 71: introduce EnDA as abbreviation here

line 267: indicated instead of indicted

line 293: called instead of "are called"

line 355: "DA coupling" instead of "DA coupled"

line 386: FESOM-ECHAM instead of FEMOS-ECHAM

Figure 1 caption: "user-provided" instead of "used-provided"

Figure 6: relative time should not have units of [s]

––––––––––––––––––––––––––––––

---

## Author Comment (AC1) · 26 Nov 2019

**Dear Dr. Kerkweg, actually, we prepared our original manuscript according to Editorial version 1.1. The new version 1.2 was published just one week before our submission, and we didn't note this update when we submitted the manuscript. Please see below for how we adapt now to the new requirements.**

Dear authors, in my role as Executive editor of GMD, I would like to bring to your attention our Editorial version 1.2:

[Figure]

https://www.geosci-model-dev.net/12/2215/2019/

This highlights some requirements of papers published in GMD, which is also available on the GMD website in the 'Manuscript Types' section:
http://www.geoscientific-model-development.net/submission/manuscript_types.html

In particular, please note that for your paper, the following requirement has not been met in the Discussions paper:
"The main paper must give the model name and version number (or other unique identifier) in the title." Please provide the version number for PDAF in the title of your revised manuscript.

**Response: Actually, the focus of the manuscript is on the model binding for AWI-CM with PDAF and not about PDAF itself (discussing all features of PDAF would be a different manuscript). To this end we now provide the version number as "AWI-CM-PDAF 1.0". However, in general the scope of the paper is wider as AWI-CM is only used as an example.**

Additionally, please note, that GMD is encouraging authors to provide a persistent access to the exact version of the source code used for the model version presented in the paper. As explained in https://www.geoscientific-modeldevelopment.net/about/manuscript_types.html the preferred reference to this release is through the use of a DOI which then can be cited in the paper. For projects in GitHub a DOI for a released code version can easily be created using Zenodo, see https://guides.github.com/activities/citable-code/ for details.

**Response: The code availability section was revised. For model binding for**

[Figure]

**AWI-CM with PDAF, which is the main focus of the manuscript is now available using Zenodo (http://doi.org/10.5281/zenodo.3551667). Actually, the ECHAM-component of AWI-CM underlies license restrictions, so that we cannot make it readily available and hence cannot provide a DOI for it. Also, PDAF is distributed via its own web site.**

Finally note, that according to our new Editorial (v1.2) all data and analysis / plotting scripts should be made available.

**Response: We now provide the plotting scripts and the output data that is used to generate the timing plots in Figs. 5 and 6 on Zenodo (http://doi.org/10.5281/zenodo.3551675)**

---

## Author Comment (AC2) · 26 Nov 2019

**We like to thank for reviewer for the careful review. Please see our response below.**

The paper describes the implementation of the software tool PDAF to a coupled ocean-atmosphere model. It discusses essentially the general structure of the PDAF software and how the coupling can be realized on a distributed computing architecture with MPI. While this is interesting, my main issues with this manuscript are the following 4 points:

1. No actual results of the assimilation system are presented. Only the execution time for different settings. It is unclear to me what the role of a reviewer can be in this case. I rather think that the paper should also include the results of such model (see also the following point).

**Response: The manuscript was prepared for the particular scope of the Journal Geoscientific Model Development (GMD) as a technical development study. As such the manuscript focuses on the technical aspects and the scalability. Discussing actual data assimilation results would not be in line with the scope. Actually, given that coupled data assimilation is a young approach, and still challenging, we think that GMD would not be the right journal to discuss application results of coupled data assimilation as we would not reach the intended readers. Apart from this, the scalability was only assessed with short experiments over 10 days (i.e. 10 analysis cycles). During this time, the assimilation process is in the initial spin-up phase and the assimilation effect is still small. Significantly longer experiments over a few months or a year would be required to get significant assimilation results. Given limited computing resources, we cannot perform full-length scalability experiments.
To respond to the authors recommendation, we have revised the introduction to better point out the status and challenges of weakly and strongly coupled data assimilation. This should clarify why here we only discuss experiments with weakly coupled assimilation.**

2. The manuscript mentions different approaches to implement the assimilation in a coupled system: in a combined state vector spanning the atmospheric and ocean model or separately. The question about which approach is better is still open and it should not be too difficult to the authors to check both approaches. This would help

also to address the previous point and add substantially to the scientific value of this paper.

**Response: While technically the strongly-coupled data assimilation is not too difficult, as is actually discussed in the manuscript, the application as such is. Strongly-coupled data assimilation is a very young approach and there are hardly any papers on this topic. By now, we know that the plain application of strongly coupled data assimilation does likely not give optimal results. To this end, we didn't attempt strongly-coupled DA in this manuscript as the assimilation results are most likely not representative. This led to the decision to discuss the model binding AWI-CM-PDAF version 1.0 for weakly coupled data assimilation. This scope is now better clarified in the manuscript.**

3. The different time scales of ocean and atmosphere are not discussed and the assimilation is done only in the ocean. To really appreciate the effectiveness of the coupling, data should be assimilated in both the atmosphere and the ocean and the question regarding the assimilation frequency should be addressed. As usual, the models should be validated against independent observations.

**Response: As mentioned before, following the scope of GMD, the manuscript discusses the PDAF model binding for a coupled model as a technical development paper. Discussing application results and the question of the assimilation frequency would be a different study, which would certainly not suited for GMD.**

4. There is too much overlap between this manuscript and previous manuscripts by the same author concerning the description of PDAF (in particular the memory coupling, general API structure). I think the author should focus this paper on the
coupling aspect and just reference to elements already published before.

**Response: We have revised Section 3 (in particular 3.1 and 3.2) to also discuss the particularities of coupling PDAF to the coupled model with multiple executables. For completeness of the manuscript, we prefer to keep aspects like the in-memory coupling or the added subroutines in the manuscript, even though quite a bit of these aspects were already discussed in the previous study (Nerger and Hiller 2012). Even more, aspects like the routine 'Assimilate_PDAF' are new, and its discussion is only possible when also discussing the routines 'Init_parallel_PDAF' and 'Init_PDAF'.**

I therefore recommend major revision before this article is published in GMD.

Minor comments:

line 46: tranDAsfers -> transfer

**Response: corrected**

page 6: MPI Communicators: is this discussion not too technical?

**Response: Given that the manuscript is submitted to GMD, we think that the degree of technicality is just right (e.g. see also Kurtz et al., 2016). The configuration of the communicators is actually a core part that makes the online coupled of PDAF with AWI-CM work.**

Section 5: How the system scales for a fixed ensemble size?

**Response: For a fixed ensemble size, the scalability is determined by the scalability of the models (as discussed in our previous papers on PDAF). As this holds likewise for assimilation with uncoupled and coupled models, we didn't perform systematic scalability tests on this aspect. We now mention the scalability for a fixed ensemble size in Sec. 4.1.**

Figure 6: the label mentions relative execution times, but the unit on the axis is [s].

**Response: corrected**

---

## Author Comment (AC3) · 26 Nov 2019

**Dear Dr. Liu,**
**thank you for commenting on our manuscript. Your comments helped to clarify some aspects in the manuscript. Please see our explanations below.**

Dear authors, Thanks a lot for making this article of the latest improvement of PDAF online on GMD for discussion.

I am Li Liu from Tsinghua University, China, leading the development of C-Coupler that is a Chinese coupler family for Earth system modelling. I am very interested in

the software framework for ensemble data assimilation, even leading a research in this topic. So I am very interested in your PDAF work, and have learned a lot from its documentations, source codes, and this article.

After reading this article, I have the following concerns:
1. After downloading and then reading the latest available code version of PDAF from your website, I guess that it does not fully include the implementation for this article.

**Response: In deed the model binding for AWI-CM which is described in the manuscript is not yet in the release package of PDAF. We will include it in the next release. To fulfill the requirements at GMD, we now provide the model binding code also at zenodo.org (http://doi.org/10.5281/zenodo.3551667).**

2. Figure 3 of this article and the source code of PDAF may indicate that the certain order of processes in the MPI_COMM_WORLD among ensemble members of the coupled model as well as its component models is required, and different members of the same component model must have the same number of processes. For example, the IDs of processes of atm_member1, ocn_member1, atm_member2, ocn_member2, atm_member3 and ocn_member3, and the processes not involved in ensemble data assimilation, must be in an ascending/descending order. Is there any restriction about the processes not involved in ensemble data assimilation?

**Response: Actually, the configuration of the parallelization is performed in the routine init_parallel_pdaf. The routine provides one common configuration, which we have found to be compatible with all models we have worked with so far. However, init_parallel_pdaf is also intended to be a template which can be adapted by the user (as is stated in the header of the file). The template**

Interactive
comment

character of the file is also the reason why its name does not start with PDAF_
and why it is not part of the PDAF library itself. Thus, while the provided default
configuration assumes a particular order of the processes, one can adapt
init_parallel_pdaf for cases with different orders. In fact, we had to adapt the
single-program variant of the routine, which is provided in the PDAF release
package, for the coupled model with 2 executables. Section 3.3 describes this
configuration.

3. Regarding P9L261_270, it seems unclear how to split the communicator
for a set of processes exclusive from ensemble data assimilation when splitting
MPI_COMM_WORLD into a group of COMM_CPLMOD. Is there any new modification
in the APIs or input files of PDAF for this functionality?

**Response: As described above, the only required change is in the routine
init_parallel_pdaf. One required change was to account for the fact the the
number of processes per model task is the number of processes for ECHAM
and those for FESOM. These are read from a namelist file. However, no changes
to the API of PDAF were required. Further, as PDAF itself is a software library,
inputs are defined by the user and hence PDAF itself does not use input files.**

4. Regarding the weakly-coupled application mentioned in P10L285_L288, it seems
unclear how to generate separate COMM_FILTER for ECHAM and FESOM? Regard-
ing this functionality, I guess that PDAF should know all component models of the
coupled model and the number of processes of each component model, and know
that weak coupling but not strong coupling is used. It may be interesting to know how
PDAF is extended for this kind of input.

**Response: The difference in the weakly and strongly coupled assimilation is really just the different setup of COMM_filter. To distinguish these cases, we have introduced a flag 'DA_couple_type' in the code, which allows to switch between both assimilation modes.**

5. Regarding Figure 2, it is still unclear of the code flowchart of different component models in weak coupling. For example, given that only ECHAM is involved in data assimilation but FESOM is not, it is unclear whether only ECHAM calls init_PDAF and Assimilate_PDAF, or FESOM has to call these two APIs cooperatively?

**Response: We have included in the flowchart only one sketch of the 'Model with DA extension' because the additional subroutine calls are added likewise in FESOM and ECHAM. This is now better clarified in Sec. 3.2.**

6. PDAF requires the filter to use the same parallel decomposition with the model. Is it possible to introduce challenges when integrating an existing filter that already has its own parallel decomposition that may be different from the model. For example, a land surface model generally uses a round-robin parallel decomposition for load balance in parallelization, which may be not suitable for a filter or will introduce new code development or lower efficiency to the filter.

**Response: Actually, PDAF was already applied with TerrSysMP, which includes a land surface model. The setup for this model was discussed by Kurtz et al. (2016). The assumption of the default setup of PDAF is that there is one pair of MPI communication calls between rank 0 and each other rank in COMM_COUPLE. Thus, communication patterns that would need several calls are not supported in the default setup of PDAF. In addition, always full sub-**

domains of state vectors are communicated. However, the number of processes in COMM_COUPLE does not need to be equal to the number of model tasks. For the default setup of PDAF, we have not attempted to provide a more general communication pattern in a generic form. In particular this would make the configuration phase much more difficult for users of PDAF while most users can use the default setup efficiently. Also, it should be possible to apply the filter to a round-robin distributed decomposition if all ensemble members use the same distribution. This is because the analysis step of PDAF does not assume compact sub-domains. However, for efficiency PDAF assumes that in the filter all ensemble members use the same parallel decomposition. If the default setup of PDAF is not directly usable, one could collect consistent sub-domains on the model processes and then communicate these to PDAF. Finally, one could also modify the communication pattern in the routines PDAF_get_state and PDAF_gather_ens of the PDAF library to adapt to a particular case. Please note that we have not included this detailed discussion in the manuscript, because we feel that this would be too much aimed for specialists.

---

## Author Comment (AC4) · 26 Nov 2019

**We like to thank for reviewer for the careful review. Please see our response below.**

The manuscript describes the application of the Parallel Data Assimilation Framework (PDAF) for coupled data assimilation, with a strong focus on strongly-coupled data assimilation (DA). An example implementation with a coupled atmosphere-ocean model is described in detail and the differences to a previous similar application of PDAF as well as to a similar application of the Data Assimilation Research Testbed

[Figure]

are explained and discussed.

While the presented MPI-based implementation for strongly-coupled data assimilation with PDAF is a logical extension of PDAF's approach for single-component models, it merits publication as a novel and highly relevant approach in the coupled case. This is well demonstrated by the comparison to and discussion of the implementations in Kurtz et al. 2016 and Karspeck et al. 2018.

However, the presented example of data assimilation for the coupled atmosphere-ocean model AWI-CM seems to fall short of demonstrating strongly-coupled data assimilation. Lines 322 to 330 describe a weakly-coupled assimilation system with coupled forecasts but observations of and assimilation in the ocean component only. The text explicitly states that "the assimilation update is only performed in the ocean compartment" which is confusing after sections 2.2 and 3.3 describe how the model states of ocean and atmosphere components are joined into a single state vector and how the model codes are extended to realize this technically. Presumably this experiment could have been realized with less code modifications than mentioned in the text. While even this setup with ocean-only assimilation into a coupled model demonstrates progress over data assimilation into a single-component model, the current presentation is unfortunate.

**Response: Actually, the model coupling is intended to support both weakly-coupled and strongly-coupled data assimilation. For the version 1.0 of the model binding AWI-CM-PDAF, we have focused on the realization of the weakly-coupled data assimilation. This is the case discussed in Section 4. The code modifications are actually the same for weakly- and strongly-coupled DA because in either case one needs to modify the model parallelization to enable the ensemble integration and the initialization of the ensemble. As we don't assimilate in the**

**atmosphere, one could have omitted the call to Assimilate_PDAF in ECHAM, but this is a minor difference.**
**We have now revised the manuscript to make the support for weakly- and strongly-coupled assimilation more explicit.**

I suggest that either the use of the presented example is well justified in the text and its relation to the previous sections and strongly-coupled DA is explained or that the example is extended to a strongly-coupled DA experiment. As it appears that large parts of the discussion and conclusion would still apply to a truly strongly-coupled data assimilation experiment, I would encourage the authors to aim for this way forward.

**Response: We have extended the Introduction to include a discussion on the status and challenges of weakly and strongly coupled DA. Given that strongly-coupled DA is a very young approach that is not yet fully established and weakly-coupled DA by itself has differences to DA in uncoupled models, we think that the focus on the weakly-coupled DA for the scalability experiment is sufficiently justified. In any case we expect that the scalability of the strongly coupled DA is very similar ot the case we have examined. We have extended the discussion to better point this out.**

Other minor points/typos:

line 46: transfers instead of tranDAsfers
line 71: introduce EnDA as abbreviation here
line 267: indicated instead of indicted
line 293: called instead of "are called"
line 355: "DA coupling" instead of "DA coupled"
line 386: FESOM-ECHAM instead of FEMOS-ECHAM

Figure 1 caption: "user-provided" instead of "used-provided"
Figure 6: relative time should not have units of [s]

**Response: We corrected all these minor points and typos.**
* * *

---

## Author Response (AR2)

**Response to the Editor**

Comments to the Author:

I'm sorry for the delay in the decision, but clear disagreement between reviewers (who I thank for their useful comments) is always a challenge and I've been considering what to do for a while.

Thank you for considering our manuscript so carefully. Obviously we had the impression that performing the technical standard benchmark of scalability testing would be sufficient, even more as the AWI-CM-PDAF implementation mainly serves as an example for the assimilation coupling strategy introduced in the manuscript, which should be obvious from the title of the manuscript.

While GMD is certainly an appropriate venue for the publication of papers describing methodologies for data assimilation, it is important that the methods are developed to a sufficiently advanced degree that their utility can be judged. It is not sufficient to present an idea that might (or even, that probably will) work. Quoting from the journal guidelines:

"Development and technical papers usually include a significant amount of evaluation against standard benchmarks, observations, and/or other model output as appropriate."

Merely showing that the code runs is not enough. Therefore in this instance I agree with Reviewer #1 who recommends major revision to include sufficient results from realistic applications in order that the merits of the system can be adequately judged.

We have now added an example in which we assimilate the sea surface temperature (SST) data over one year. We discuss the root mean square errors with regard to the assimilated SST data, but also with independent in situ data provided by the EN4 data set. Please note, that even before we did not just showed that 'the code runs' but, from the computing viewpoint, we carefully examined the scalability, which is a standard benchmark to evaluate compute performance. Anyway, now we also show that the assimilation is successful. Actually, there are two studies (Mu et al. (2020) and Tang et al. (2020)) that already base on the AWI-CM-PDAF code and demonstrate the successful application of data assimilation into the ocean. However, these studies don't discuss the implementation aspect of the data assimilation system, which is done in the manuscript. Given that Tang et al. (2020) also assimilate SST data (next to subsurface profile data) and discuss the effect of this assimilation onto the atmosphere we have not included this aspect into the manuscript and kept our application section to the essential discussion of root mean square errors.

Please also note that we don't consider it reasonable to include an experiment with strongly-coupled data assimilation or the assimilation of atmospheric observations into this manuscript. In the response to the reviewer we give detailed reasons for this (which partly repeat what we replied in the previous response).

**Response to Reviewer #1**

We like to thank the reviewer the comments and persistence with regard to the application results. The editor has clarified that such results should be included. To this end, we now discuss assimilation results from an experiment assimilating SST observations over one year. Please find our detailed response below.

For reference, these are the 3 first main points of my previous review (point 4 has been addressed):

1. No actual results of the assimilation system are presented. Only the execution time for different settings. It is unclear to me what the role of a reviewer can be in this case. I rather think that the paper should also include the results of such model (see also the following point).

In the new Section 5 we have now added an application example in which we assimilate the SST data over one year. We assess the assimilation results computed the root mean square errors with regard to the assimilated data (where the reduction of RMSEs is necessary) and by computing RMSEs for the EN4 data set of profile observations from UK MetOffice. This is an independent data set. The reductions of the RMSE for both temperature and salinity shows that the assimilation is successful. We also refer to two studies (Mu et al. 2020 and Tang et al. 2020) which both base on the AWI-CM-PDAF code we discuss in the manuscript. We kept the analysis of the results in Section 5 short to avoid overlaps with Tang et al. (2020), where also SST data is assimilated.

2. The manuscript mentions different approaches to implement the assimilation in a coupled system: in a combined state vector spanning the atmospheric and ocean model or separately. The question about which approach is better is still open and it should not be too difficult for the authors to check both approaches. This would help also in addressing the previous point and add substantially to the scientific value of this paper.

We hope that the reviewer agrees that to just 'check both approaches' would not be scientifically sound. In particular the very new aspect of strongly-coupled DA is a current research topic, which cannot just be answered in some subsection of a paper. What would be the value if we included an experiment with strongly-coupled DA? If the result would not show an improvement of strongly-coupled DA, this could simply be the reason of insufficient tuning. Likewise, if a spontaneous experiment shows improvements over the weakly-coupled approach, we would still not know if the result is representative because this would require careful tuning. To this end, including an experiment just for completeness might just lead to misleading impressions about strongly-coupled DA. On the other hand, a detailed study is clearly beyond a manuscript whose focus is on the technical, i.e. implementation, aspects of the assimilation system.

3. The different time scales of ocean and atmosphere are not discussed and the assimilation is done only in the ocean. To really appreciate the effectiveness of the coupling, data should be assimilated in both the atmosphere and the ocean and the question regarding the assimilation frequency should be addressed. As usual, the models should be validated against independent observations.

We fully agree that the aspect of the different time scales in the atmosphere and ocean is a relevant topic for coupled DA. Further, one of course constrains the model best when one assimilates observations of both the atmosphere and the ocean. However, there is no principal requirement to assimilate data into both the atmosphere and ocean. For example Kunii et al. (2017) only assimilate atmospheric observations, while Mu et al. (2020) only assimilate observations of the ocean and sea ice. These studies show that the assimilation into only one of the components has a scientific relevance on its own. Because of this we decided to focus the initial version of the data assimilation code AWI-CM-PDAF 1.0 onto the assimilation into the ocean. While aimed at the DA into the ocean, the paper describes how the system can be extended to the assimilation in the atmosphere and for strongly coupled DA. As also many published studies in GMD show, it is a common approach to develop a software step by step, assessing these steps scientifically, and also publish such steps. We consider the aspects of assimilating atmospheric data and of strongly-coupled DA for the future work, while the current software version 1.0 (and the DA coupling strategy, which is the main focus of the manuscript) is sufficiently mature for publication.

As described above, we now included an experiment discussing the results obtained from assimilating the SST data validated against independent observations from the EN4 data set. These results clearly show that the assimilation system works well.

The arguments from the authors, are essentially the following:

A: the authors think that it is enough for GMD to just present run times and presenting actual results is out-of-scope for GMD
B: only a short experiment (10 days) was conducted
C: it is unclear if the strong assimilation does actually work better than weak-assimilation (with our present understanding)

Please refer to the full response available at https://www.geosci-model-dev-discuss.net/gmd-2019-167/gmd-2019-167-AC2.pdf

For me, my 3 major comments are still relevant and within the scope of the journal. They have not been addressed in this review.

Even so this is a bit redundant, let us shortly comment on A-C:

Regarding A: The new section 5 now includes an application example.
Regarding B: Please see below for the reason why we did run only a set of experiments over 10 days each.
Regarding C: We explained above that a simple strongly-coupled DA experiment as suggested by the reviewer in the former comment 2 cannot yield a conclusive answer to point C. Strongly-coupled DA is a research topic that needs careful consideration to be scientifically relevant.

I have checked the 5 papers at the front page (as of 27 November, when I first read the authors reply) and all present the actual results of the model (as opposed to just run-times). Most of them also do actually compare the model with observations. Also the paper by Kurtz et al, 2016 published by GMD mentioned by the author and using PDAF, displays the results of a twin experiment (Figure 6 and Figure 9-12). I am surprised that the authors thinks that this is out of scope for the journal.

It also unclear why the authors limit their simulation to just 10 days which is very short for an ocean model at this resolution (from 160 km to 30 km). Most of the fields would remain fairly constant over that time scale (as also noted by the authors). The authors mentioned computing resources as a problem, but at the same time, they mentioned that a full year simulation should be completed in 6.5 hours (requiring 79,000 core-hours). I am left with the question why the authors tackle this problem before securing the necessary computing resources to make a meaningful simulation.

The choice for using 10 days was not motivated by insufficient computing resources. It is just common practice to perform scalability tests only over a period which is sufficiently long for consistent results. To this end, running scalability experiments over several months or a year would just be a waste of compute resources. This aspect is now more clearly described in Sec. 4.1 by writing *"The length of these experiments is chosen to be long enough so that the execution time is representative to assess the scalability. However, the assimilation effect will be rather small for these 10 analysis steps."*

Also, to me it is quite natural that when a new method is introduced or implemented one needs to show its benefit compared to other techniques.

The benefit is actually discussed in Sec. 5 in accordance with the manuscript's focus on the particular strategy to combine the data assimilation with the coupled model model: Compared to an offline-coupled DA system, the online-coupled DA system presented in the manuscript is at least 4 times faster. (Note, we didn't actually code the offline-coupled DA system, but the benefit in execution time can be quite well estimated as is done in the manuscript.) Also the system discussed in the manuscript is significantly faster than the system discussed by Karspeck et al. (2018).

The other points raised in my previous review have been addressed in an adequate manner.

My understanding of the scope of GMD might be wrong (and presenting just run times as opposed of actual results and not presenting improvements relative to other approaches is acceptable). If this is the case, then the Editor is invited to correct me.

After the clarification by the reviewer we have now added an example case assimilating SST data. Please note that already before we presented improvements relative to other approaches. However, these were aimed at compute performance rather than assimilation performance.

References:

[revised manuscript text omitted]